# Leaf Traits Explain the Growth Variation and Nitrogen Response of *Eucalyptus urophylla* × *Eucalyptus grandis* and *Dalbergia odorifera* in Mixed Culture

**DOI:** 10.3390/plants13070988

**Published:** 2024-03-29

**Authors:** Han Zhang, Yahui Lan, Chenyang Jiang, Yuhong Cui, Yaqin He, Jiazhen Deng, Mingye Lin, Shaoming Ye

**Affiliations:** 1College of Forestry, Guangxi University, Nanning 530004, China; 2009401005@st.gxu.edu.cn (H.Z.); yahui_lan@163.com (Y.L.); 2109302007@st.gxu.edu.cn (C.J.); 2109302001@st.gxu.edu.cn (Y.C.); 2109401009@st.gxu.edu.cn (Y.H.); dengjiazhen1012@163.com (J.D.); lmy1605@163.com (M.L.); 2Guangxi Key Laboratory of Forest Ecology and Conservation, Guangxi University, Nanning 530004, China

**Keywords:** leaf functional traits, mixed plantation, N application, growth, net photosynthetic rate

## Abstract

Mixed cultivation with legumes may alleviate the nitrogen (N) limitation of monoculture *Eucalyptus*. However, how leaf functional traits respond to N in mixed cultivation with legumes and how they affect tree growth are unclear. Thus, this study investigated the response of leaf functional traits of *Eucalyptus urophylla* × *Eucalyptus grandis* (*E. urophylla* × *E. grandis*) and *Dalbergia odorifera* (*D. odorifera*) to mixed culture and N application, as well as the regulatory pathways of key traits on seedling growth. In this study, a pot-controlled experiment was set up, and seedling growth indicators, leaf physiology, morphological parameters, and N content were collected and analyzed after 180 days of N application treatment. The results indicated that mixed culture improved the N absorption and photosynthetic rate of *E. urophylla* × *E. grandis*, further promoting seedling growth but inhibiting the photosynthetic process of *D. odorifera*, reducing its growth and biomass. Redundancy analysis and path analysis revealed that leaf nitrogen content, pigment content, and photosynthesis-related physiological indicators were the traits most directly related to seedling growth and biomass accumulation, with the net photosynthetic rate explaining 50.9% and 55.8% of the variation in growth indicators for *E. urophylla* × *E. grandis* and *D. odorifera*, respectively. Additionally, leaf morphological traits are related to the trade-off strategy exhibited by *E. urophylla* × *E. grandis* and *D. odorifera* based on N competition. This study demonstrated that physiological traits related to photosynthesis are reliable predictors of N nutrition and tree growth in mixed stands, while leaf morphological traits reflect the resource trade-off strategies of different tree species.

## 1. Introduction

As part of the plant economic spectrum, leaf functional traits are defined as the morphological/structural, chemical, and physiological characteristics of leaves that play a key role in the whole life cycle of plants [1,2]. These traits essentially reflect a plant’s trade-off strategy between the cost of leaf construction and photosynthetic output; therefore, these traits reflect the plant’s economics [3]. Regarding leaf construction, leaf area (LA), leaf thickness (LT), leaf volume (LV), leaf fresh mass (LFM), and leaf dry mass (LDM) are the basic leaf traits that directly reflect leaf construction strategies [4,5,6]. For example, a larger LA represents a larger area for light capture but also means more demand for resources such as water and nutrients [4]. LFM, LDM, and leaf mass fraction (LMF) were assessed at the leaf level and individual level for plant investment in leaf tissues and organs, respectively [7,8]. Moreover, the demand for light drives an increase in the light capture area per unit biomass (i.e., specific leaf area, SLA), which contributes to leaf photosynthesis and improved N use efficiency [9,10,11]. Leaf tissue density (LTD) and leaf dry matter content (LDMC) are used to estimate the compactness level and extent of leaf tissue growth, which may be related to plant responses to soil nutrient resources [6,12]. Regarding output, the net photosynthetic rate (Pn) is a decisive indicator of photosynthetic efficiency in leaves and is influenced by stomatal conductance (gs), transpiration rate (Tr), and intercellular carbon dioxide concentration (Ci) [13,14], while chlorophyll content (Chl) determines the flux size of photons received in leaves [15]. In addition, Rubisco carboxylation efficiency (CE) reflects the carboxylation efficiency of the key rate-limiting enzyme of photosynthesis [16], and leaf water-use efficiency (WUE) reflects the water use by the leaves and is closely related to the photosynthetic process [17]. On this basis, the plant fast–slow economic spectrum theory proposes that differences in plant traits exhibited along a resource gradient are key to the successful survival of different species [1]. That is, fast-growing species adapt to resource-rich environments and build simpler leaves using fewer resources with looser structures but higher N contents and photosynthetic rates, while slow-growing species build more complex leaves using more resources with denser structures but lower N contents to adapt to barren conditions [18]. However, additional research is needed to supplement our knowledge of leaf functional traits in response to N.

N is an essential, massive element that limits plant growth and development and plays an important role in leaf construction and production. Studies have shown that the chloroplast N content can reach 75% of the total leaf N content (LNC), and 30–40% of this N is allocated to ribulose-1,5-bisphosphate carboxylase/oxygenase (Rubisco), whose main role is to assimilate CO_2_ and is a rate-limiting enzyme for photosynthesis [19,20]. Evans and Clarke (2019) allocated leaf N according to function, indicating that the N associated with photosynthesis accounts for 54% of the total leaf N content, while the remaining N is used for the construction of the nucleus, cell wall, mitochondria, cytoplasm, and other parts of the plant [21]. Recent studies have shown that N addition increases the total N content and SLA and decreases the LDMC in the leaves of two herbaceous species, resulting in a preference for a “quick investment-return” survival strategy [22]. For herbaceous plants, N addition has an asymmetric effect on changes in leaf functional traits and biomass [23]. Recent studies of woody plants have shown that changes in the functional traits of tree leaves (e.g., SLA and LDMC) are correlated with soil N dynamics [24,25,26]. Therefore, changes in the functional traits of plant leaves may be good predictors of environmental N availability.

*Eucalyptus* species are timber species that are widely planted worldwide, and their advantages of rapid growth and high productivity compensate for the shortage of quality timber and slow the deforestation of natural forests [27]. However, the current multigenerational succession management model used for pure *Eucalyptus* stands places greater demands and challenges on soil fertility as well as the environment [28,29], and land degradation limits the growth of *Eucalyptus* [29,30]. In particular, the availability of N in the soil is one of the main factors limiting the productivity of agroforestry [31]. N fertilization can increase productivity to a certain extent. However, excessive N application can accelerate environmental pollution [32,33]. Recent studies have indicated that intercropping *Eucalyptus* plants with leguminous tree species can improve plant nitrogen use efficiency and the potential for ecological service functions [34,35,36]. In mixed systems where nitrogen-fixing plants are present or under fertilization conditions, plant leaves are the direct beneficiaries of improved N use, and the response of leaf functional traits to improved N conditions will directly determine plant “gain” [37,38]. However, the adaptive changes in the functional leaf traits of these two plant species in mixed systems and their impact on plant growth are unknown.

Based on this information, this study used controlled pot experiments of *Eucalyptus urophylla* × *eucalyptus grandis* (*E. urophylla* × *E. grandis*) and *Dalbergia odorifera* (*D. odorifera*) to investigate the effects of mixed cultivation, N addition, and co-treatments on leaf functional traits as well as further on the growth and biomass of seedlings. In this study, leaf morphological/structural, physiological, and chemical traits related to plant growth and development were collected according to previous methods (Appendix A) [39,40,41,42,43,44,45,46]. The objectives of this research were (1) to understand the adaptive mechanisms of leaf traits in *E. urophylla* × *E. grandis* and *D. odorifera* in response to mixed cultivation and N application, (2) to test the hypothesis that mixed plantation and N application and their cotreatments promote the growth and development of *E. urophylla* × *E. grandis* and *D. odorifera*, and (3) to elucidate the correlation between key leaf traits and the growth of *E. urophylla* × *E. grandis* and *D. odorifera*, as well as their respective regulatory pathways.

## 2. Results

### 2.1. Responses of Leaf Physiological Traits to Mixed Planting and N Application

With increasing N application, there was a trend toward significantly increasing LNC, Chl a, Chl b, Chl a+b, Car, Pn, gs, photosynthetic N use sufficiency (PNUE), WUE, and CE, and decreasing Ci in the monoculture *E. urophylla* × *E. grandis*. Only mixed culture had no obvious effect on the LNC, Chl a, Chl b, Chl a+b, Car, gs, Tr, Ci, or CE of *E. urophylla* × *E. grandis* (Figure 1). Cotreatments of mixed cultivation and N addition significantly increased the LNC, Pn, gs, PNUE, WUE, and CE of *E. urophylla* × *E. grandis* (Figure 1A,F,G,J–L). However, the Chl a, Chl b, Chl a+b, Car, and Tr of *E. urophylla* × *E. grandis* increased significantly only under the higher N addition treatments (N2 (6 g urea pot^−1^) and N3 (12 g urea pot^−1^)) in the mixed plantation (Figure 1B–E,H). Moreover, cotreatment with mixed culture and N fertilization significantly reduced the Ci of the leaves of *E. urophylla* × *E. grandis* (Figure 1I).

For *D. odorifera*, the mixed plantation treatment significantly reduced the leaf Pn, PNUE, and WUE (Figure 2F,J,K). The high N addition treatments (N2 and N3) significantly increased the LNC, Chl b, Pn, gs, Tr, PNUE, and CE but decreased the Ci and WUE (Figure 2A,C,F–L). The cotreatment of mixed planting and different N applications significantly increased LNC (N1 (3 g urea pot^−1^) and N3), gs (N2 and N3), Tr (N1, N2, and N3), and Ci (N2) and reduced the Chl a (N2), Chl a+b (N2), Pn (N1, N2, and N3), PNUE (N1, N2, and N3), WUE (N1, N2 and N3), and CE (N1, N2, and N3) of *D. odorifera* (Figure 2).

### 2.2. Responses of Leaf Morphological Traits to Mixed Planting and N Application

Compared to monoculture, mixed cultivation significantly increased the LA, LDM, LDMC, and LTD of *E. urophylla* × *E. grandis*, while notably reducing the SLA (*p* < 0.05). On the other hand, N addition increased the LA, LDM, LDMC, and LTD of *E. urophylla* × *E. grandis*, while reducing its LT, LV, LFM, SLA, and LMF (*p* < 0.05). Under the combined treatment of low N addition (CK and N1) and mixed cultivation, the LA, LV, LFM, LDM, LDMC, and LTD of *E. urophylla* × *E. grandis* were all higher than those under the combined treatment of low N addition (CK and N1) and monoculture. Conversely, the aforementioned traits under the combined treatment of high N addition (N2 and N3) and mixed cultivation were unchanged or reduced compared to the monoculture treatment. In addition, compared to monoculture, low N addition treatments (CK and N1) reduced the SLA of *E. urophylla* × *E. grandis* leaves under mixed cultivation, while high N treatments (N2 and N3) increased the SLA. Notably, the LMF of *E. urophylla* × *E. grandis* under both monoculture and mixed cultivation showed a decreasing trend with increasing N application (Figure 3).

Compared to monoculture, mixed cultivation treatment significantly reduced the LA, LT, LV, LFM, LDM, LDMC, and LMF of *D. odorifera* under all nitrogen application treatments but significantly increased the SLA and LTD (*p* < 0.05). The overall leaf morphological traits of *D. odorifera* were insensitive to N addition, and only the LMF decreased with increasing N application rate (Figure 4).

### 2.3. Response of Tree Species Growth to Mixed Plantation and N Addition

Compared to monoculture, the mixed plantation significantly promoted plant height, diameter, branch number, and biomass of leaf, branch, stem, root, shoot, and total plant of *E. urophylla* × *E. grandis* (Figure 5). Different N application treatments significantly increased the plant height, diameter, branch number, and biomass of leaf, branch, stem, root, shoot, and whole plant but reduced the root/shoot of *E. urophylla* × *E. grandis* under monoculture treatment (Figure 5A–J). Under the cotreatment of mixed cultivation and N application, plant height, diameter, branch number, and the biomass of leaf, branch, stem, root, shoot, and total plant of *E. urophylla* × *E. grandis* were significantly greater than those under the other treatments (Figure 5A–I). The PLER of *E. urophylla* × *E. grandis* was greater than 0.5, but N application reduced the PLER (Figure 5K). The N1 and N2 treatments significantly increased the CR_ED_ of *E. urophylla* × *E. grandis* (Figure 5L).

N application obviously increased the height, diameter, branch number, and biomass of the leaves, rachises, stems, shoots, roots, and whole plants of *D. odorifera* in the monoculture treatment (*p* < 0.05). Mixed cultivation did not significantly affect the height or branch number or the biomass of the leaf, leaf rachis, stem, shoot, root, or whole plant of *D. odorifera* but did significantly reduce the diameter (Figure 6A–I). Under different nitrogen application treatments, mixed cultivation significantly (*p* < 0.05) reduced height, ground diameter, branch number, and biomass of leaf, rachis, stem, shoot, root, and plant of *D. odorifera* compared to monoculture (Figure 6A–I). The PLER and CR_DE_ of *D. odorifera* were lower than 0.5 and 1, respectively, and the N1 and N2 treatments significantly reduced the PLER and CR_DE_ (Figure 6K,L).

### 2.4. Relationships between Leaf Functional Traits and Their Contributions to Plant Growth

Under monoculture treatment, there was a significant positive correlation between Pn, gs, Tr, WUE, CE, LNC, PNUE, Chl a, Chl b, Chl a+b, and Car of *E. urophylla* × *E. grandis* (0.395 < R < 0.990, *p* < 0.05). Yet, they were all significantly negatively correlated with Ci (−0.948 < R < −0.623, *p* < 0.01). Regarding leaf morphological traits, the LA of *E. urophylla* × *E. grandis* showed a significant positive correlation with LV (R = 0.533, *p* = 0.015) and a negative correlation with LT (R = −0.527, *p* = 0.017). The SLA was significantly negatively correlated with LT, LFM, LDM, and LTD (−0.798 < R < −0.471, *p* < 0.05). And LDMC was significantly positively correlated with LDM and LTD (R = 0.689 and 0.731, *p* = 0.001 and 0.000). Pn, Tr, CE, and LNC showed significant negative correlations with LT, LFM, and LMF (−0.739 < R < −0.467, *p* < 0.05), but significant positive correlations with LDMC (0.512 < R < 0.611, *p* < 0.05). Additionally, Chl a, Chl b, Chl a+b, and Car exhibited significant negative correlations with LV, LFM, and LMF (−0.748 < R < −0.443, *p* < 0.05) (Figure 7A). Under mixed cultivation conditions, significant positive correlations are observed among Pn, gs, Tr, WUE, CE, and LNC (0.451 < R < 0.965, *p* < 0.05). Additionally, leaf pigments display highly significant positive correlations (0.847 < R < 0.958, *p* < 0.01). Furthermore, significant positive relationships are also evident between Pn, WUE, CE, LNC, and the pigments Chl b, Chl a+b, and Car (0. 473 < R < 0.623, *p* < 0.05). Regarding leaf morphological traits, significant positive correlations were found among LT, LV, LFM, LDM, LTD, and LMF of *E. urophylla* × *E. grandis* (0. 466 < R < 0.935, *p* < 0.05), while all exhibited significant negative correlations with SLA (−0.887 < R < −0.603, *p* < 0.01). Moreover, there was a general negative correlation between the morphological and physiological traits of *E. urophylla* × *E. grandis*, except for SLA and Ci (Figure 7B). Mantel’s test results indicate that the growth indices of monoculture *E. urophylla* × *E. grandis* are highly significantly correlated with Pn, Ci, WUE, CE, LNC, PNUE, and Car (R > 0.4, *p* < 0.01), and are significantly correlated with gs, Chl a, Chl a+b, LDMC, and LTD (0.2 < R < 0.4, *p* < 0.05) (Figure 7A). For mixed-cultivated *E. urophylla* × *E. grandis*, leaf Pn, Ci, WUE, CE, and LNC all show highly significant correlations with growth indices (R > 0.4, *p* < 0.01), as does Tr (0.2 < R < 0.4, *p* < 0.01). The gs also exhibits a significant correlation with growth indices (0.2 < R < 0.4, *p* < 0.05) (Figure 7B). The results of the RDA indicated that Pn, LMF, Tr, Chl a, and PNUE significantly influenced the growth of *E. urophylla* × *E. grandis*, explaining 50.9%, 9.6%, 4.1%, 2.8%, and 2.4%, respectively, of the variation in growth indices (Figure 8A).

In the monoculture of *D. odorifera*, there were significantly positive correlations between Pn, gs, Tr, CE, LNC, and PNUE (0.612 < R < 0.980, *p* < 0.01), but all were negatively correlated with Ci and WUE (−0.849 < R < −0.446, *p* < 0.05). The SLA had strong negative correlations with LFM, LDM, LDMC, and LTD (−0.759 < R < −0.541, *p* < 0.05). Furthermore, the correlation between leaf physiological and morphological traits is relatively weak and not statistically significant (−0.4 < R < 0.4, *p* > 0.05) (Figure 7A). Under mixed cultivation treatment, leaf Pn, WUE, CE, and PNUE of *D. odorifera* are all highly significantly positively correlated (0.828 < R < 0.990, *p* < 0.001). The LNC and the contents of Chl a and Chl b exhibit significant correlations (R = 0.546 and 0.495, *p* < 0.05). Significant positive correlations are observed among the LA, LV, LFM, and LDM of *D. odorifera* (0.831 < R < 0.944, *p* < 0.001). Correlations between the physiological and morphological traits of *D. odorifera* leaves from mixed culture were generally weak, but there were strong associations between LDMC and leaf Pn, CE, and PNUE (−0.525 < R < −0.498, *p* < 0.05) (Figure 7D). In addition, Mantel’s test showed significant associations between *D. odorifera* growth indicators and Pn, gs, Tr, WUE, and PNUE in the monoculture treatment (R > 0.2, *p* < 0.05), while these leaf traits were only SLA, Pn, and CE in the mixed cultivation treatment (0.2 < R < 0.4, *p* < 0.05) (Figure 7C,D). According to the RDA results, the growth of *D. odorifera* was significantly affected by Pn, Tr, PNUE, and Chl a+b, which explained 55.8%, 5.3%, 2.4%, and 2.4%, respectively, of the variation in the growth indices (Figure 8B).

### 2.5. Potential Pathways of Leaf Trait Regulation of Plant Growth

According to the PLS-SEM results, the growth indices and biomass of *E. urophylla* × *E. grandis* and *D. odorifera* are jointly regulated by multiple leaf traits (Figure 9). Specifically, the growth (height, diameter, and branch number) of *E. urophylla* × *E. grandis* was directly regulated by LNC (0.616), leaf physiological traits (0.453), and leaf pigment content (−0.239), while its biomass (leaf, branch, stem, and root) was directly positively influenced by treatment (0.548) and leaf physiological traits (0.432). Overall, the growth of *E. urophylla* × *E. grandis* was positively regulated by treatment, LNC, and leaf physiological traits but negatively regulated by leaf morphological traits (Figure 9A). Treatment (0.275) and leaf physiological indicators (1.260) had direct positive impacts on the growth (height, diameter, and branch number) of *D. odorifera*, while leaf pigment content (−0.282) directly negatively regulated its growth. Moreover, the biomass (leaf, branch, stem, and root) of *D. odorifera* was directly positively regulated by treatment (0.320) and leaf physiological traits (1.075). Overall, LNC, leaf physiological traits, and leaf pigment content had positive regulatory effects on *D. odorifera*, while treatment and leaf morphological traits had overall negative impacts on its growth (Figure 9B).

## 3. Discussion

### 3.1. Adaptation of Leaf Traits of E. urophylla × E. grandis and D. odorifera to Mixed Plantation and N Addition Treatments

In general, variation in the functional traits of plant leaves is considered an adaptive change in response to the external environment [47,48]. In the present study, the variation in leaf morphological, physiological, and nutritional traits of *E. urophylla* × *E. grandis* and *D. odorifera* in response to mixed and N treatments was investigated (Figure 1, Figure 2, Figure 3 and Figure 4). The results suggest that the two species have different adaptation strategies.

N addition has been shown to increase the LNC based on mass [49,50]. The results of the present study showed that mixed culture alone did not significantly alter the LNC of *E. urophylla* × *E. grandis*, whereas N application alone significantly increased the LNC of *E. urophylla* × *E. grandis* (Figure 1A), indicating that soil N availability is the main driver of LNC. In mixed systems, plant acquisition of N depends on the competitive ability of the species [51,52]. Obviously, the interspecific competition in mixed cultivation tilted the balance of resource acquisition compared to the relatively equal intraspecific competition of the same plant species in monoculture. A higher root biomass allows for a greater soil N uptake capacity in *E. urophylla* × *E. grandis* [53], even robbing part of the N in *D. odorifera* to achieve interspecific N transfer [54,55]. Regarding leaf physiological traits, the Pn in *E. urophylla* × *E. grandis* under monoculture increased with N application, and this change was accompanied by an increase in gs and CE and a decrease in Ci (Figure 1). This result is consistent with most N addition experiments [56,57], indicating that N application alleviated the N limitation of photosynthesis. Correlation analysis revealed that N application increased the LNC and thus significantly increased leaf photosynthesis (Figure 7). An important reason is that the increase in leaf N promotes chlorophyll synthesis (Figure 8A) and improves light use efficiency [58,59]. Second, the increase in Pn benefited from the increase in CE (Figure 5), which improved the CO_2_ conversion efficiency [44,46]. In addition, N addition may indirectly improve photosynthetic efficiency by improving leaf stomatal permeability (gs), WUE, and photoprotective capacity (by Car) [60,61,62]. In the present study, the mixed treatment and the interaction with N application increased the photosynthesis of *E. urophylla* × *E. grandis* leaves to different degrees while significantly decreasing the photosynthetic rate of *D. odorifera* (Figure 1 and Figure 2). A reason for this finding is that the significant reduction in nonstomatal factors such as WUE and CE in *D. odorifera* leaves grown in mixed cultivation limits the photosynthetic response process [63]. Moreover, the significant decrease in chlorophyll and carotenoid contents in *D. odorifera* plants grown in a mixed cultivation system reduced the light capture and photoprotective capacity of the leaves [43]. The combined effects of mixed culture and N application increased the water and nutrient competitive ability of *E. urophylla* × *E. grandis* and decreased that of *D. odorifera*, which resulted in the overall improvement and decline of the leaf photosynthetic physiological functions of *E. urophylla* × *E. grandis* and *D. odorifera*, respectively [64,65].

Leaf morphological traits are important for the regulation of environmental adaptation in plants. Different plants can achieve similar resource acquisition capacities in both above- and below-ground parts in two ways: one way is to obtain more leaf biomass but less root biomass at lower SLAs, and the other way is to gain less leaf biomass but more root biomass at higher SLAs [66,67]. The results indicate that mixed cultivation of *E. urophylla* × *E. grandis* and *D. odorifera* tends to favor the second method of resource acquisition compared to monoculture. This means that mixed cultivation promotes the allocation of more root biomass for nutrient uptake by *E. urophylla* × *E. grandis* and *D. odorifera* (Figure 5 and Figure 6J) while reducing the material used for leaf construction, thereby increasing the light capture area per unit dry mass (Figure 3 and Figure 4). Takigahira and Yamawo (2019) suggested that this strategy may be linked to interspecific competition between *E. urophylla* × *E. grandis* and *D. odorifera* in mixed cultivation [68]. Moreover, some studies have shown that LMF increases with nutrient availability but decreases with light [69]. In this study, a noteworthy phenomenon was the decrease in the LMF of *E. urophylla* × *E. grandis* and *D. odorifera* with increasing N application under both monoculture and mixed cultivation, which was contrary to the results of Zou et al. [66], who found that N application increased the LMF of *Machilus pauhoi* seedlings. The results of this study suggest that the LMF under mixed cultivation decreased with increasing N application. This difference may be due to the improved leaf light conditions in the mixed system, which depend on the increased light harvesting capacity resulting from increased height [70]. Generally, it is believed that a high SLA and a low LDMC represent rapid nutrient acquisition and promote plant growth in fertile soils, while a low SLA and a high LDMC are commonly found in plants in poor environments [22,71]. It is clear that SLA and LDMC had opposite responses to N gradients in *E. urophylla* × *E. grandis* and *D. odorifera*, although this trend was weaker for *D. odorifera* (Figure 3 and Figure 4). This finding suggested that *E. urophylla* × *E. grandis* and *D. odorifera* may have employed different resource acquisition strategies under monoculture. N addition decreased the SLA of monocultured *E. urophylla* × *E. grandis* and increased LDMC, which may be due to N causing an imbalance of N and P in the trees, leading to a phosphorus limitation [24]; another possibility is that intraspecific N competition in *E. urophylla* × *E. grandis* resulted in a conservative resource acquisition strategy, which better explains how the N3 treatment alleviated the N limitation in this study [68]. Furthermore, mixed culture changed the response trends of SLA and LDMC to N gradients in *E. urophylla* × *E. grandis* and *D. odorifera* (Figure 3 and Figure 4). These findings suggested that mixed cultivation and N application enhance the resource availability of *E. urophylla* × *E. grandis* and reduce the resource accessibility of *D. odorifera*. This may be due to N intensifying interspecific competition and reducing the competitiveness of *D. odorifera* [72]. In addition, our results showed that cotreatment of mixed plantations and N application reduced the LA, LT, and LV of *D. odorifera* while increasing the LTD, indicating that N treatment induced changes in *D. odorifera* resistance under mixed culture conditions [73,74].

Understanding the relationships between plant leaf traits is necessary for understanding plant coordination strategies, but the effect of mixed stands on these relationships is still unclear [20]. Our results showed that *E. urophylla* × *E. grandis* and *D. odorifera* leaf morphological traits and physiological traits were negatively correlated overall (Figure 7). There appears to be a contradiction in nitrogen allocation between photosynthesis and leaf construction, indicating that plants have trade-offs between leaf investment and output [75]. Mixed planting did not change the strong positive correlation between the leaf physiological traits of *E. urophylla* × *E. grandis* but significantly weakened the correlation between the leaf physiological traits of *D. odorifera* (Figure 7), indicating that mixed planting caused photosynthetic physiological disorders in *D. odorifera* [76,77]. In addition, the correlation between the leaf morphological traits of *E. urophylla* × *E. grandis* was enhanced in the mixed plantation, which optimized leaf function and might be an important manifestation of improving the competitiveness of *E. urophylla* × *E. grandis* [68,78,79].

### 3.2. Competition for N Promotes the Growth of E. urophylla × E. grandis and Inhibits the Growth of D. odorifera

It has been shown that mixed culture has major productivity advantages [54,80]. However, this study showed that mixed cultivation had opposite effects on the growth of *E. urophylla* × *E. grandis* and *D. odorifera*. Compared to that in the monoculture system, the significant increase in aboveground biomass of *E. urophylla* × *E. grandis* in the mixed system contributed the majority (PLER > 0.5), but the aboveground biomass of *D. odorifera* in the mixed system decreased (PLER < 0.5) compared to that in the monoculture system. This result is also supported by the data on plant height, ground diameter, branch number, and plant biomass (Figure 5 and Figure 6). On the one hand, this difference appears to be the result of differences in the ability to use light, space, and soil resources among different tree species [81,82], and these differences result in reduced or facilitated competitiveness among tree species in mixed systems [83]. In this study, *E. urophylla* × *E. grandis* exhibited a taller plant height (Figure 5A), a greater number of branches (Figure 5E), and greater root biomass (Figure 5H). These characteristics allowed for improved light, water, and fertilizer access; enhanced competitiveness (Figure 5L); and promoted growth. On the other hand, interspecific relationships based on complementary effects are largely influenced by N availability [84]. Yao et al. reported that leguminous plants transfer N to eucalyptus trees through root contact, as shown by the ^15^N isotope labeling method [35,54]. This phenomenon has been observed in mixed plant species such as wheat and faba bean [85], maize and soybean [86], and *Acacia mangium* and *Eucalyptus* [55]. This enhances the N uptake of *Eucalyptus* plants and fully utilizes the N fixation ability of leguminous plants, improving the N utilization efficiency of the entire mixed system [87]. Additionally, the mixed system may redistribute N between different species through mycorrhizal mediation, which could be a significant factor in the differences in growth between *Eucalyptus* and leguminous plants [88,89].

Plants need adequate N for growth and development. This study revealed that N fertilization effectively improved soil N availability, resulting in increased height, diameter, branch number, and biomass in both the monoculture *E. urophylla* × *E. grandis* and *D. odorifera* (Figure 5 and Figure 6). Notably, compared to the control without N application, N fertilization reduced the above-ground biomass of the entire mixed cultivation system. This was primarily due to the significant decrease in the aboveground biomass of the mixed *D. odorifera* plants with N addition (Figure 6G). This result is consistent with the stress gradient hypothesis (SGH), which supports the shift from facilitation to competition in plant interactions as survival pressure decreases [90]. Compared to those in the monocultures with the same N application rate, the higher root/shoot ratios in the mixed *E. urophylla* × *E. grandis* and *D. odorifera* treatment groups likely indicated that more biomass was allocated to the roots to compete for N (Figure 5 and Figure 6J). Moreover, the competition index (CR_ED_) for *E. urophylla* × *E. grandis* was much greater than the CR_DE_ for *D. odorifera* in the mixed treatment (Figure 5 and Figure 6L), indicating that *E. urophylla* × *E. grandis* is more competitive. Additionally, research has shown that when forest growth conditions improve beyond the stress gradient range, the complementarity of certain species combinations may increase [91]. This may explain the decrease in total biomass for mixed *E. urophylla* × *E. grandis* and the increase in biomass for *D. odorifera* under the N3 treatment (Figure 5 and Figure 6).

### 3.3. Growth-Related Leaf Traits and Possible Regulatory Pathways in E. urophylla × E. grandis and D. odorifera

According to the classical “leaf economic spectrum” theory, leaves mediate the effects of heterogeneous environments on tree growth and development [3,18,92]. This study revealed several leaf functional traits significantly associated with the growth of *E. urophylla* × *E. grandis* and *D. odorifera*. Mantel tests revealed significant correlations between leaf traits LNC, Pn, gs, Ci, WUE, and CE and growth indicators under both pure and mixed cultivation of *E. urophylla* × *E. grandis* (Figure 7A,B). RDA also revealed that Pn, LMF, Tr, Chl a, and PNUE significantly influenced the growth and biomass accumulation of *E. urophylla* × *E. grandis* (Figure 8A). This indicates that the growth of *E. urophylla* × *E. grandis* is strongly dependent on leaf N supply and related photosynthetic processes. These traits were significantly promoted by mixed culture and N application, thereby promoting the height, ground diameter, branching, and biomass of *E. urophylla* × *E. grandis* (Figure 8A). Leaf morphological traits have been reported to be related to plant adaptation to environmental changes [40,46]. In this study, the LMF was found to be significantly negatively correlated with the growth indices and biomass of *E. urophylla* × *E. grandis* (Figure 8A). According to the optimal allocation theory, the reduction in light resources under monoculture cultivation induced an increase in *E. urophylla* × *E. grandis* LMF to mitigate light competition pressure by enhancing potential resource absorption capacity [47]. However, in mixed cultivation, the light resources available for *E. urophylla* × *E. grandis* may have improved, promoting plant growth (Figure 8A) [48]. Furthermore, structural equation modeling indicated that LNC, leaf pigment content, and other leaf physiological traits directly affect the growth of *E. urophylla* × *E. grandis*, but leaf morphological traits may indirectly influence the growth of *E. urophylla* × *E. grandis* by directly negatively regulating LNC and leaf pigment content (*p* < 0.05).

For *D. odorifera*, whether in monoculture or mixed cultivation, the Pn was still the most important leaf trait affecting plant growth (Figure 7C,D and Figure 8B). Furthermore, the RDA indicated that Tr, PNUE, and Chl a+b were significantly negatively correlated with biomass in *D. odorifera*, consistent with the results for *E. urophylla* × *E. grandis*. However, mixed cultivation reduced these indicators, which inhibited the growth of *D. odorifera* (Figure 9B). Moreover, we discovered that physiological indicators related to photosynthesis (Pn, gs, Tr, Ci, WUE, CE, and PNUE) directly regulate the growth (1.260) and biomass (1.075) of *D. odorifera* and have the greatest overall effect (Figure 9B). Therefore, a decrease in leaf photosynthetic efficiency is likely the cause of the inhibition of *D. odorifera* growth in mixed cultivation. In addition, for both *E. urophylla* × *E. grandis* and *D. odorifera*, the leaf N content, pigment content, and other physiological traits all had opposite effects on plant growth compared to leaf morphological traits (Figure 9). This discovery confirms the trade-off between leaf construction and output, which also determines plant growth [47].

## 4. Materials and Methods

### 4.1. Trees and Soil

In this experiment, three-month-old *E. urophylla* × *E. grandis* (mean height: 44.62 cm) and one-year-old *D. odorifera* (mean height: 47.68 cm) plants were used because they had similar heights. These seedlings were obtained from the Guangxi Bagui Seedling Company (Nanning, Guangxi, China). The soil was collected at the Guangxi Gaofeng State-Owned Forest Farm. The acidic red soil in the 0–30 cm layer was air-dried and subsequently sieved through a 1 cm mesh to eliminate coarse impurities. The soil pH was 4.81, and the N content was 0.675 g kg^−1^. To ensure soil ventilation, perlite was added to the soil, and the substrates were mixed well (soil–perlite = 13:2 (*v*:*v*)). Nonwoven seedling bags (r = 25 cm and h = 45 cm) were used for culture containers, and each pot was filled with approximately 55 kg of soil (Figure 10).

### 4.2. Experimental Design

The cultivation experiment was conducted from March to September 2021 at the nursery of Guangxi University, College of Forestry (22°51′4.8″ N, 108°17′30.3″ E). During cultivation, the air temperature variation ranged from 25–40 °C, and the humidity ranged from 50–80%. The sunshine duration of the whole year and the cultivation period were 1654.7 h and 937 h, respectively. The experiment was conducted in a randomized group design with three species combination patterns (*E. urophylla* × *E. grandis* monoculture, *D. odorifera* monoculture, and mixed culture of *E. urophylla* × *E. grandis* and *D. odorifera*), and two seedlings were planted in each pot (Figure 10). In each cultivation pattern, four N application levels (no urea application (CK), 3 g urea pot^−1^ (N1), 6 g urea pot^−1^ (N2), and 12 g urea pot^−1^ (N3)) were set separately. In total, this study was set up with 16 treatments (2 species × 2 cultivation patterns × 4 N addition levels), and each treatment contained 5 replicates. Urea (CH_4_N_2_O) was used as an artificial nitrogen source and applied to the soil as a solution 1 week after tree planting. Thereafter, each pot was given an equal amount of water each day to ensure the growth of the plants.

### 4.3. Leaf Functional Traits

After 180 days of N treatment, the Pn, gs, Tr, and Ci of the mature leaves of *E. urophylla* × *E. grandis* and *D. odorifera* were measured using a Li-6800 portable photosynthesis system (LI-COR, Lincoln, NE, USA) from 9:00 to 11:00 on a sunny day. According to a previous study, the light intensity of the artificial light source was set at 1200 μmol m^−2^ s^−1^ (*E. urophylla* × *E. grandis*) and 1000 μmol m^−2^ s^−1^ (*D. odorifera*), and the ambient CO_2_ concentration was 380 μmol (CO_2_) mol^−1^. The leaf chamber temperature was 31.3 ± 0.6 °C, and the ambient atmospheric pressure was 99.97 ± 0.02 kPa. WUE and CE were calculated by the ratios of Pn to Tr and Pn to Ci, respectively [63]. Photosynthetic N use sufficiency (PNUE) was calculated as the ratio of the net photosynthetic rate to the leaf nitrogen concentration. The leaf pigments were extracted using an 80% acetone solution, and the absorbances of the extracts at 470, 646, and 663 nm were measured using a Libra S22 UV/Vis spectrophotometer (Biochrom Ltd., Cambridge, UK). Leaf chlorophyll a (Chl a), chlorophyll b (Chl b), chlorophyll a+b (Chl a+b), and carotenoid (Car) contents were calculated using the equations deduced by Lichtenthaler [93].

Additionally, approximately 2400 leaves were collected from the upper, middle, and lower parts of the plants in the 12 treatments. To prevent errors caused by leaf water dissipation, the LT and LFM of each leaf were measured immediately after the leaf was removed. The leaf thickness was measured three times along the main vein using a spiral micrometer with an accuracy of 0.01 mm, and the mean value was considered the leaf thickness. The leaves were scanned using a scanner, and subsequently, the LA was determined through the use of the image processing package (Fiji) in ImageJ2 software (v1.53t National Institutes of Health, Bethesda, MD, USA). Finally, the LDM was determined using the same electronic balance after drying to a constant weight at 60 °C. LV was obtained by multiplying LA by LT; SLA was calculated by the ratio of LA to LDM [94]; LDMC was expressed by the ratio of LDM to LFM; and LTD was calculated by dividing LDM by LV [12]. From the whole-leaf perspective, the LMF at the branching level was expressed as the ratio of total leaf dry weight to total dry weight of leaves and branches [40]. Finally, the leaf samples were digested with H_2_SO_4_-H_2_O_2_, and the total N content of the leaves was determined through micro-Kjeldahl methods.

### 4.4. Plant Growth and Biomass

Plant height, ground diameter, and branch number were measured for each plant in the different treatments 180 days after N application. Plant height was defined as the vertical distance from the ground to the point of growth of the main stem. The ground diameter was defined as the diameter of the main stem 10 cm from the ground. Leaves, stems, branches, primary roots, and lateral roots of *E. urophylla* × *E. grandis* and *D. odorifera* seedlings from each treatment were separated 180 days after the N application. The plant samples were fixed at 95 °C for 2 h and then heated at 60 °C to a constant weight, after which the results were recorded. The shoot biomass was calculated by adding the leaf, branch, and stem biomass of the seedlings, and the root/shoot ratio is the ratio of seedling root biomass to aboveground biomass.

### 4.5. Statistics and Analysis

The partial land equivalent ratio (PLER) was calculated with reference to the following equation [95]:(1)$$\mathrm{PLER}=B_{mix}/B_{mo}$$
where B_mix_ and B_mo_ represent the aboveground biomass (g plot^−1^) under mixed stand and monoculture cultivation, respectively.

The competitive ratio (CR) was calculated based on the following equation [96]:(2)$${\mathrm{CR}}_{\mathrm{ED}}=B_{\mathrm{ME}}/{(B}_{E}\times Z_{\mathrm{ME}})÷B_{\mathrm{MD}}/\left(B_{D}\times Z_{\mathrm{MD}}\right)$$
(3)$${\mathrm{CR}}_{DE}=B_{\mathrm{MD}}/{(B}_{D}\times Z_{\mathrm{MD}})÷B_{\mathrm{ME}}/{(B}_{E}\times Z_{\mathrm{ME}})$$
where CR_ED_ indicates the competitive ratio of *E. urophylla* × *E. grandis* relative to that of *D. odorifera* and CR_DE_ indicates the competitive ratio of *D. odorifera* relative to that of *E. urophylla* × *E. grandis*. B_ME_ and B_E_ indicate the biomass of the whole plant (g plot^−1^) of *Eucalyptus* in mixed systems and monocultures, respectively; B_MD_ and B_D_ indicate the biomass of the whole plant (g plot^−1^) of *D. odorifera* in mixed stands and monoculture, respectively; and Z_ME_ and Z_MD_ indicate the proportion of plants in mixed cultivation (Z_ME_ = 0.5 and Z_MD_ = 0.5) of *E. urophylla* × *E. grandis* and *D. odorifera*, respectively. The capital letters “E” and “D” are used to represent *E. urophylla* × *E. grandis* and *D. odorifera*, respectively, in this study.

To explore the effects of different treatments and their interactions on plants, the data from the experiment were subjected to one-way and two-way ANOVA with a significance test (Duncan’s test, *p* < 0.05) using SPSS version 19.0^®^ (SPSS Corp., Chicago, IL, USA). The Mantel test in R (version 4.3.1) and redundancy analysis (RDA) in Canoco 5.0 (Wageningen University and Research, Wageningen, The Netherlands) were used to predict the relationships between leaf functional traits and biomass in *E. urophylla* × *E. grandis* and *D. odorifera*. The potential correlation between leaf physiological and morphological traits was explored using correlation analysis in R (version 4.3.1). In addition, partial least squares structural equation modeling (PLS-SEM) was used to analyze the regulatory effects of leaf traits on plant growth. SmartPLS 4.0 (SmartPLS GmbH, Inc., Oststeinbek, Germany) was used to construct formative indicator models. A path-based weighting scheme with a maximum of 1000 iterations was chosen. We evaluated the significance of the paths between each latent variable using significance tests and adjusted for nonsignificant paths. Finally, we evaluated the structural models using the coefficient of determination (R^2^), effect size (f^2^), and cross-validation redundancy (Q^2^). Chin (1998) used R^2^ to measure the correlation between the explained variance of a latent variable and its total variance [97]. High, medium, and low explanations are denoted by R^2^ values of 0.670, 0.333, and 0.190, respectively. f^2^ was used to evaluate the structural equation modeling for the degree of influence of each pathway [98]. The predictive relevance of the structural model was evaluated using the nonparametric Stone–Geisser test, and Q^2^ was used to determine the predictive relevance of the model for a given structure [99,100].

## 5. Conclusions

This study investigated the N response and growth characteristics of *E. urophylla* × *E. grandis* and legume tree species on mixed plantations based on leaf functional traits. The results of multiple analyses indicated that the differences in nutrient acquisition characteristics and N competitive ability of the tree species themselves led to different responses of *E. urophylla* × *E. grandis* and *D. odorifera* leaf N content, physiological traits, and morphological traits to N application and mixed cultivation. This ultimately promoted *E. urophylla* × *E. grandis* growth but inhibited *D. odorifera* growth under the combined treatment of mixed cultivation and N application. Notably, leaf N content and physiological indicators related to photosynthesis directly determine seedling growth, while leaf morphological traits participate in the trade-off between seedling resource acquisition and indirect effects on seedling growth. The results of this study can help to understand the N utilization characteristics of trees in mixed stands and ultimately provide a reference for fertilization decisions and the selection of mixed tree species in the forestry production process.

## Figures and Tables

**Figure 1 plants-13-00988-f001:**
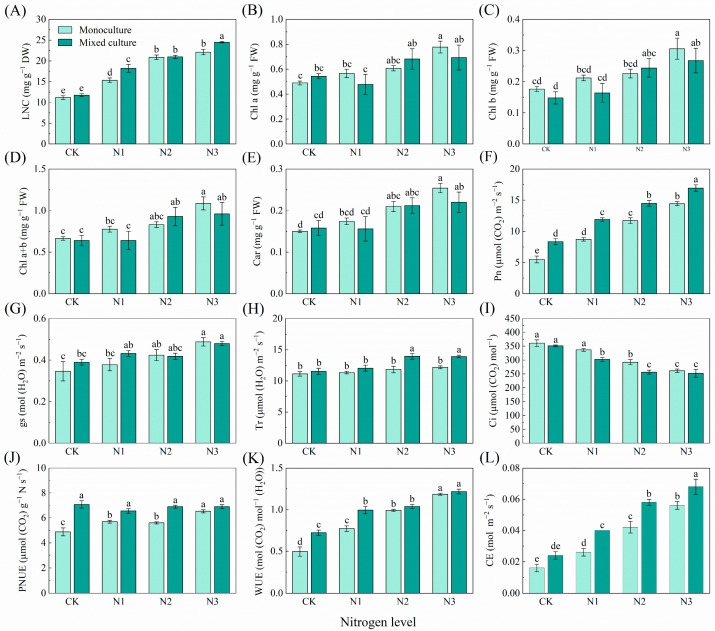
Effects of mixed planting and N addition on the physiological traits of *E. urophylla* × *E. grandis* seedlings. The differences in (**A**) leaf nitrogen content (LNC), (**B**) chlorophyll a (Chl a), (**C**) chlorophyll b (Chl b), (**D**) total chlorophyll content (Chl a+b), (**E**) carotenoid (Car), (**F**) net photosynthetic rate (Pn), (**G**) stomatal conductance (gs), (**H**) transpiration rate (Tr), (**I**) intercellular CO_2_ concentration (Ci), (**J**) photosynthetic nitrogen use sufficiency (PNUE), (**K**) water-use efficiency (WUE), and (**L**) carboxylation efficiency of Rubisco (CE) in *E. urophylla* × *E. grandis* were compared via one-way ANOVA. Significant differences (*p* < 0.05; *n* = 5) between treatments are indicated by different lowercase letters.

**Figure 2 plants-13-00988-f002:**
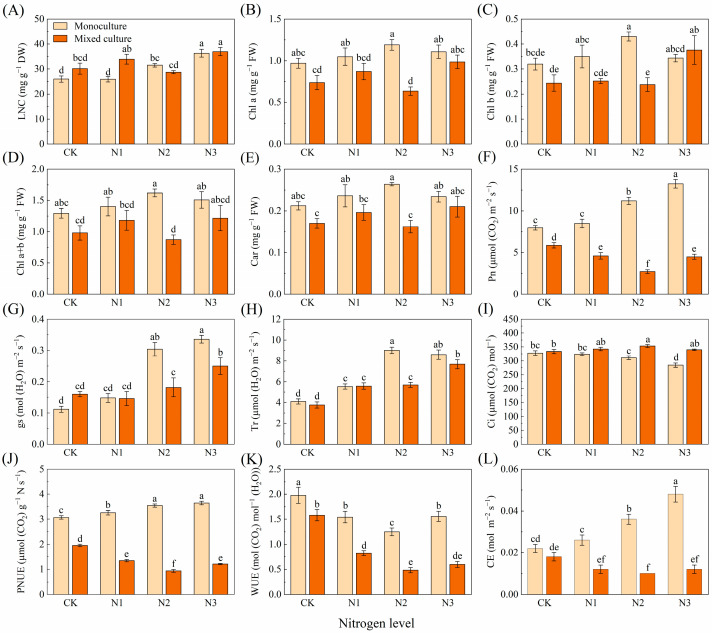
Effects of mixed planting and N addition on the physiological traits of *D. odorifera* seedlings. The differences in (**A**) leaf nitrogen content (LNC), (**B**) chlorophyll a (Chl a), (**C**) chlorophyll b (Chl b), (**D**) total chlorophyll content (Chl a+b), (**E**) carotenoid (Car), (**F**) net photosynthetic rate (Pn), (**G**) stomatal conductance (gs), (**H**) transpiration rate (Tr), (**I**) intercellular CO_2_ concentration (Ci), (**J**) photosynthetic nitrogen use sufficiency (PNUE), (**K**) water-use efficiency (WUE), and (**L**) carboxylation efficiency of Rubisco (CE) in *D. odorifera* were compared via one-way ANOVA. Significant differences (*p* < 0.05; *n* = 5) between treatments are indicated by different lowercase letters.

**Figure 3 plants-13-00988-f003:**
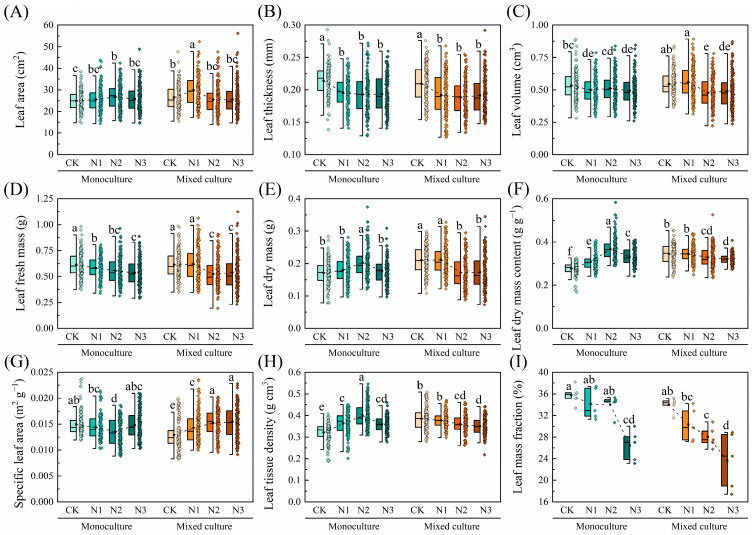
Effects of mixed cultivation and N addition on leaf morphological traits of *E. urophylla* × *E. grandis*. (**A**) leaf area (LA), (**B**) leaf thickness (LT), (**C**) leaf volume (LV), (**D**) leaf fresh mass (LFM), (**E**) leaf dry mass (LDM), (**F**) leaf dry mass content (LDMC), (**G**) specific leaf area (SLA), (**H**) leaf tissue density (LTD), and (**I**) leaf mass fraction (LMF). Dashed lines indicate trends in the mean values of leaf traits with different N application treatments. Different lowercase letters indicate significant differences between different treatment combinations (*p* < 0.05; *n* = 50).

**Figure 4 plants-13-00988-f004:**
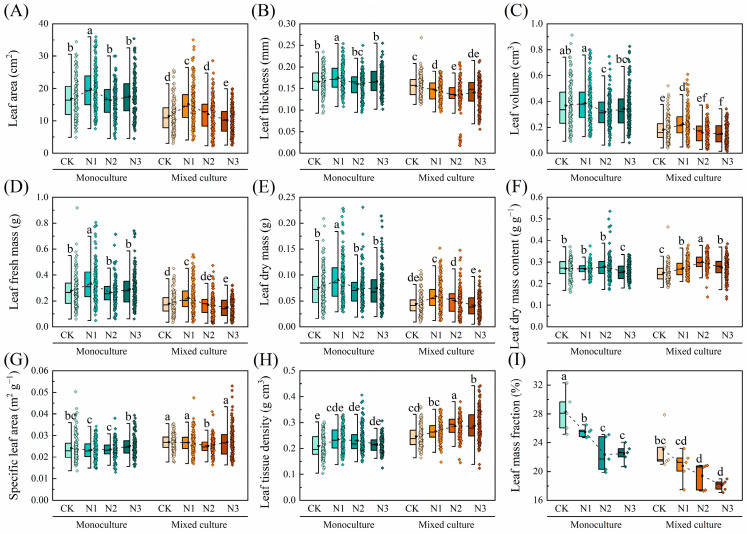
Effects of mixed cultivation and N addition on the leaf morphological traits of *D. odorifera*. (**A**) leaf area (LA), (**B**) leaf thickness (LT), (**C**) leaf volume (LV), (**D**) leaf fresh mass (LFM), (**E**) leaf dry mass (LDM), (**F**) leaf dry mass content (LDMC), (**G**) specific leaf area (SLA), (**H**) leaf tissue density (LTD), and (**I**) leaf mass fraction (LMF). Dashed lines indicate trends in the mean values of leaf traits with different N application treatments. Different lowercase letters indicate significant differences between different treatment combinations (*p* < 0.05; *n* = 50).

**Figure 5 plants-13-00988-f005:**
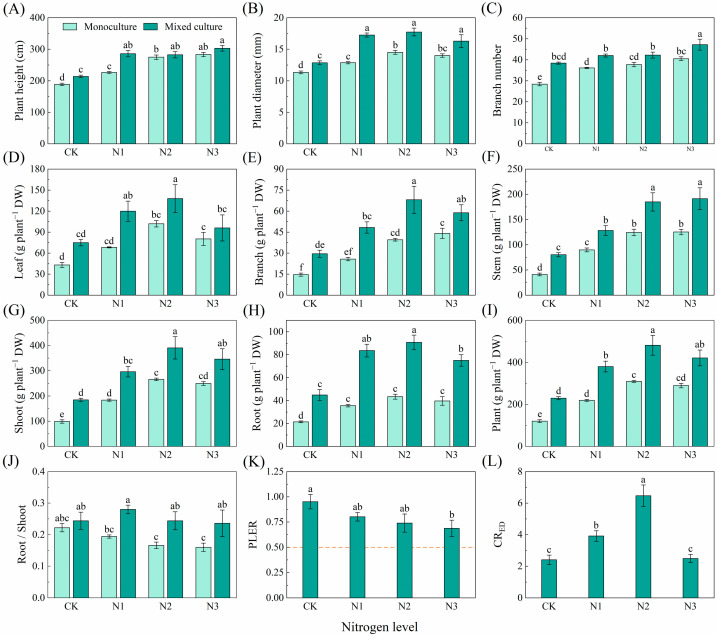
Effects of mixed planting and N addition on the growth of *E. urophylla* × *E. grandis*. The differences in (**A**) plant height, (**B**) diameter, (**C**) branch number, biomass of (**D**) leaf, (**E**) branch, (**F**) stem, (**G**) shoot (gs), (**H**) root, (**I**) whole plant, (**J**) root/shoot, (**K**) PLER, and (**L**) CR_ED_ were compared via one-way ANOVA. The partial land equivalent ratio (PLER) was calculated based on plant aboveground parts (**K**), and the competitive ratio (CR_ED_) was calculated based on the total biomass (**L**). The shoot biomass of monoculture and mixed cultivation seedlings are equal at the dotted line in subfigure (**K**). Different lowercase letters indicate significant differences in the data between different treatments (*p* < 0.05; *n* = 5).

**Figure 6 plants-13-00988-f006:**
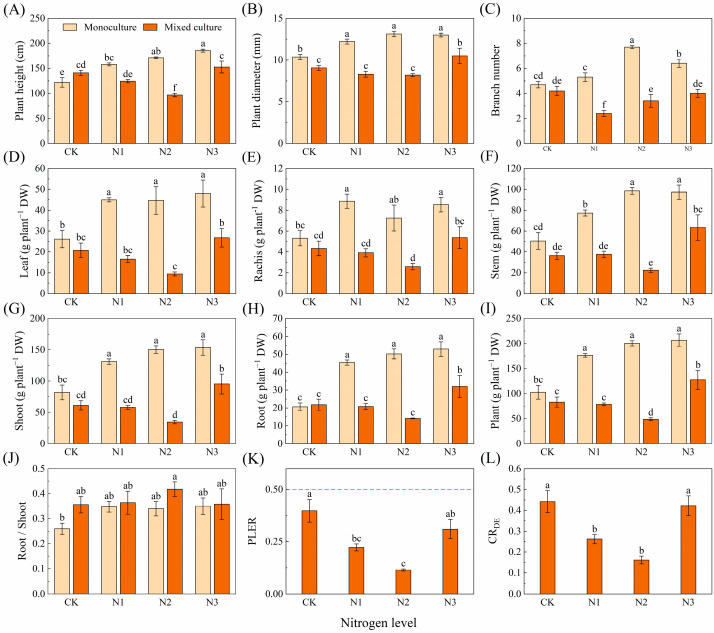
Effects of mixed planting and N addition on the growth of *D. odorifera*. The differences in (**A**) plant height, (**B**) diameter, (**C**) branch number, biomass of (**D**) leaf, (**E**) branch, (**F**) stem, (**G**) shoot (gs), (**H**) root and (**I**) whole plant, (**J**) root/shoot, (**K**) PLER, and (**L**) CR_ED_ were compared via one-way ANOVA. The partial land equivalent ratio (PLER) was calculated based on plant aboveground parts (**K**), and competitive ratios (CR_DE_) were calculated based on the total biomass (**L**). The shoot biomass of monoculture and mixed cultivation seedlings are equal at the dotted line in subfigure (**K**). Different lowercase letters indicate significant differences in the data between different treatments (*p* < 0.05; *n* = 5).

**Figure 7 plants-13-00988-f007:**
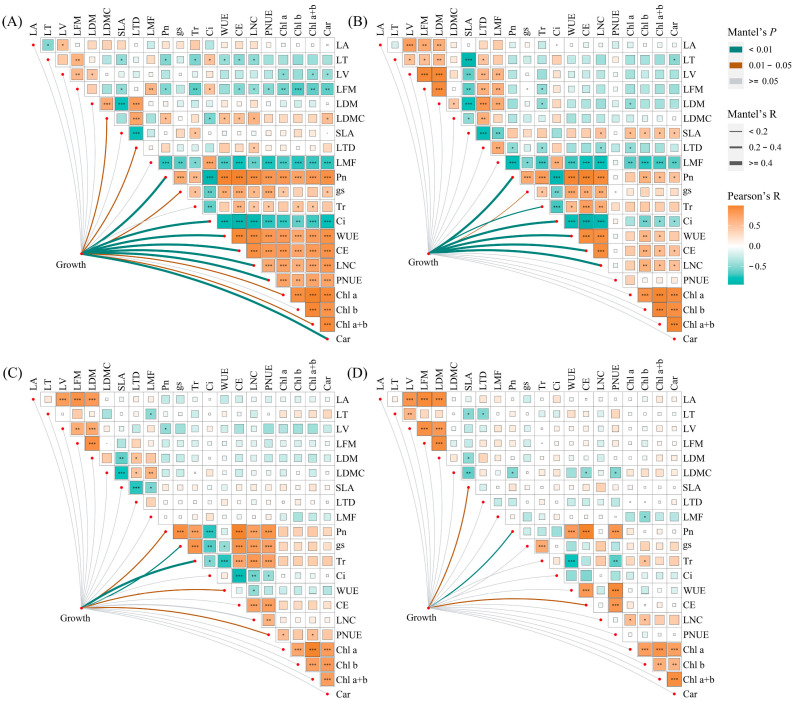
Correlations between leaf functional traits and Mantel test results for the relationships between leaf functional traits and growth. (**A**) *E. urophylla* × *E. grandis* in monoculture; (**B**) *E. urophylla* × *E. grandis* in mixed culture; (**C**) *D*. *odorifera* in monoculture; and (**D**) *D*. *odorifera* in mixed culture. The growth variables included plant height, ground diameter, branch number, leaf, branch, stem, root, shoot, whole plant biomass, and root/shoot. * Significant at *p* < 0.05; ** Significant at *p* < 0.01; *** Significant at *p* < 0.001.

**Figure 8 plants-13-00988-f008:**
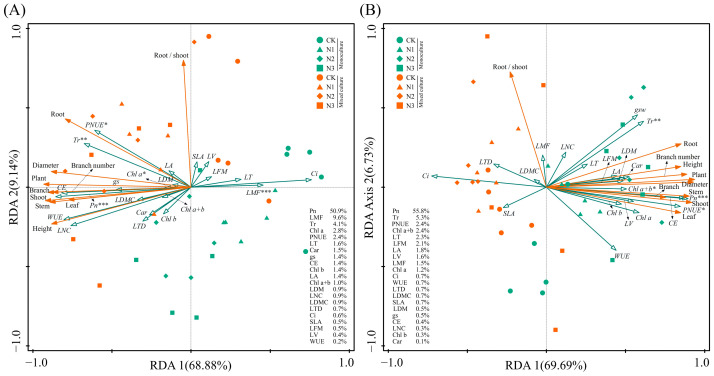
Redundancy analysis showing the effects of leaf functional traits on the growth indices of *E. urophylla* × *E. grandis* (**A**) and *D. odorifera* (**B**). *, **, and *** indicate significant differences at the 0.05, 0.01, and 0.001 levels, respectively.

**Figure 9 plants-13-00988-f009:**
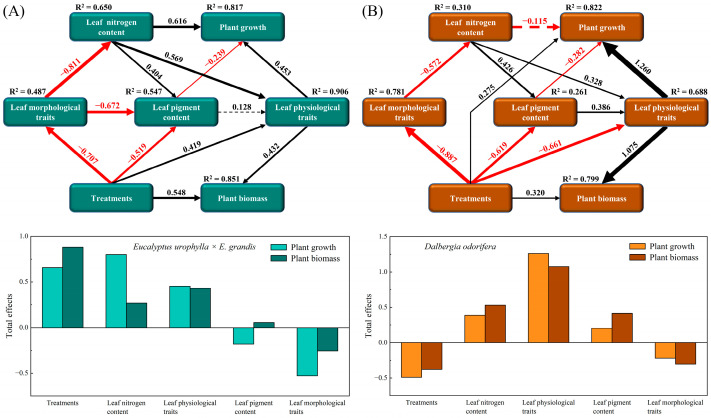
Cascading relationships between plant growth and leaf functional traits. Partial least squares path modeling was employed to disentangle the major pathways by which different treatments influenced leaf nitrogen content; leaf morphological traits (LA, LT, LV, LFM, LDM, LDMC, SLA, LTD, and LMF); leaf physiological traits (Pn, gs, Tr, WUE, CE, and PNUE); leaf pigment contents (Chl a, Chl b, and Car); plant growth (height, diameter, and branch number); and plant biomass (leaf, branch, stem, and root) in *E. urophylla* × *E. grandis* (**A**) and *D*. *odorifera* (**B**). The solid black and red arrows represent positive and negative significant relationships, respectively, while the dashed black and red arrows represent positive and negative nonsignificant relationships, respectively. The numbers on the paths indicate the correlation coefficients, and the thickness of the arrows indicates the strength of the correlation.

**Figure 10 plants-13-00988-f010:**
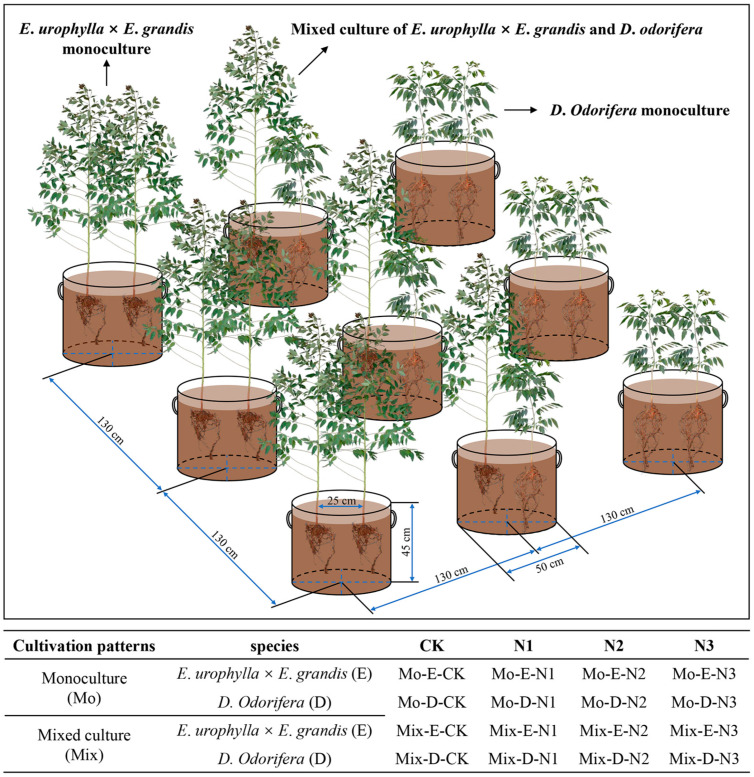
Diagram of the mixed plantation, monoculture, and experimental treatments.

## Data Availability

All the data generated or analyzed during this study are included in this published article (and its Appendix A).

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
