# Peer review of "Leaf Traits Explain the Growth Variation and Nitrogen Response of Eucalyptus urophylla × Eucalyptus grandis and Dalbergia odorifera in Mixed Culture"

_plants, 2024, doi:10.3390/plants13070988_

Round 1

Reviewer 1 Report

Comments and Suggestions for Authors

In In the abstract part, the authors first provide background, and aim and then directly move towards results. Please systematically arrange the abstract to summarize the manuscript such as background, aims, methods, results and conclusion.

Line 92: “and their interactions on the leaf functional traits of..” confusing statement please revise this sentence.

216-217: “Moreover, there was a general negative correlation between the morphological and physiological traits of E. urophylla × E. grandis, except for SLA and Ci (Figure. 7B)” . Please provide the reason for the negative correlation between morpho and physiological traits in the discussion section.

Line 202-249: “2.4. Relationships among leaf functional traits and their contributions to plant growth” Please provide r or r2 values, t-values, p-values and standard error in a separate table obtained from the multiple correlation tests in the supplementary file. And then put r-values along with p-values within the text to identify the positive/negative relationship.

Line 496: Please follow the journal guidelines for in-text citations.

Line 506: Same as above

Line 510: Same as above

Line 527 Same as above

Figure 8: Why do the authors perform RDA rather than PCA?

Overall the figures are scientifically robust and help to better understand the results.

Line 543: “redundancy analysis (RDA) was used…” Through which software the authors perform RDA?

545: “The potential correlation between leaf physiological and morphological traits was explored using correlation analysis”. Using which software please clarify.

Line 562: “This study is our first attempt” Please avoid colloquial terminology.

Line 572. “These 572 research results”…. Same as above

Comments on the Quality of English Language

Minor editing of English language required

Author Response

Dear reviewers,

Thank you for offering us an opportunity to improve the quality of our submitted manuscript (plants-2859534). We appreciated very much the constructive and insightful comments provided by the academic editor and reviewers. In this revision, we have addressed all of these comments/suggestions. We hope the revised manuscript has now met the publication standard of your journal.

The revised portions are marked in blue ink in the paper.

Below, our point-by-point responses to the issues raised by the reviewers are listed in blue text.

Response to the comments of Reviewer #1

In the abstract part, the authors first provide background, and aim and then directly move towards results. Please systematically arrange the abstract to summarize the manuscript such as background, aims, methods, results and conclusion.

Response:

Thank you for your reasonable suggestions, which greatly improved the quality of our manuscript. We have added the methodological content of this study in the abstract part (Lines 15-17).

Correction:

In this study, pot experiment was set up and seedling growth indicators, leaf physiology, morphological parameters and N content were collected and analyzed after 180 days of N application treatment.

Line 92: “and their interactions on the leaf functional traits of..” confusing statement please revise this sentence.

Response: Thank you, we've revised that sentence (Lines 94-97).

Correction:

Based on this information, this study used controlled pot experiments of Eucalyptus urophylla × eucalyptus grandis (E. urophylla × E. grandis) and Dalbergia odorifera (D. odorifera) to investigate the effects of mixed cultivation, N addition and co-treatments on leaf functional traits as well as further on growth and biomass of seedlings.

Lines 216-217: “Moreover, there was a general negative correlation between the morphological and physiological traits of E. urophylla × E. grandis, except for SLA and Ci (Figure. 7B)”. Please provide the reason for the negative correlation between morpho and physiological traits in the discussion section.

Response: Thank you for your suggestion. We have added possible reasons for the negative correlation between leaf morphological and physiological traits in the discussion of the manuscript (Lines 388-392).

Correction:

Our results showed that E. urophylla × E. grandis and D. odorifera leaf morphological traits and physiological traits were negatively correlated overall (Figure. 7). There appears to be a contradiction in nitrogen allocation between photosynthesis and leaf construction, indicating that plants have trade-offs between leaf investment and output [75].

Line 202-249: “2.4. Relationships among leaf functional traits and their contributions to plant growth” Please provide r or r2 values, t-values, p-values and standard error in a separate table obtained from the multiple correlation tests in the supplementary file. And then put r-values along with p-values within the text to identify the positive/negative relationship.

Response: Thank you for your constructive suggestions. We have added the r-values, p-values, t-values, and standard errors for the correlation analysis of leaf trait of E. urophylla × E. grandis and D. odorifera under different cultivation patterns to the supplementary file (Table S3-10). In addition, we have revised the results section according to your suggestions, in the hope that the modified version will enhance your understanding of the manuscript (Lines 216-266).

Correction:

Under monoculture treatment, there was a significant positive correlation between Pn, gs, Tr, WUE, CE, LNC, PNUE, Chl a, Chl b, Chl a+b, and Car of E. urophylla × E. grandis (0.395 < R < 0.990, P < 0.05). Yet, they were all significantly negatively correlated with Ci (-0.948 < R < -0.623, P < 0.01). Regarding leaf morphological traits, the LA of E. urophylla × E. grandis showed a significant positive correlation with LV (R = 0.533, P =0.015) and a negative correlation with LT (R = -0.527, P =0.017). The SLA was significantly negatively correlated with LT, LFM, LDM and LTD (-0.798 < R < -0.471, P < 0.05). And, LDMC was significantly positively correlated with LDM and LTD (R = 0.689 and 0.731, P =0.001 and 0.000). Pn, Tr, CE, and LNC showed significant negative correlations with LT, LFM, and LMF (-0.739 < R < -0.467, P < 0.05), but significant positive correlations with LDMC (0.512 < R < 0.611, P < 0.05). Additionally, Chl a, Chl b, Chl a+b, and Car exhibited significant negative correlations with LV, LFM, and LMF (-0.748 < R < -0.443, P < 0.05) (Figure. 7A). Under mixed cultivation conditions, significant positive correlations are observed among Pn, gs, Tr, WUE, CE, and LNC (0.451 < R < 0.965, P < 0.05). Additionally, leaf pigments display highly significant positive correlations (0.847 < R < 0.958, P < 0.01). Furthermore, significant positive relationships are also evident between Pn, WUE, CE, LNC, and the pigments Chl b, Chl a+b, and Car (0. 473 < R < 0.623, P < 0.05). Regarding leaf morphological traits, significant positive correlations were found among LT, LV, LFM, LDM, LTD, and LMF of E. urophylla × E. grandis (0. 466 < R < 0.935, P < 0.05), while all exhibited significant negative correlations with SLA (-0.887 < R < -0.603, P < 0.01). Moreover, there was a general negative correlation between the morphological and physiological traits of E. urophylla × E. grandis, except for SLA and Ci (Figure. 7B). Mantel’s test results indicate that the growth indices of monoculture E. urophylla × E. grandis are highly significantly correlated with Pn, Ci, WUE, CE, LNC, PNUE, and Car (R > 0.4, P < 0.01), and are significantly correlated with gs, Chl a, Chl a+b, LDMC, and LTD (0.2 < R < 0.4, P < 0.05) (Figure. 7A). For mixed-cultivated E. urophylla × E. grandis, leaf Pn, Ci, WUE, CE, and LNC all show highly significant correlations with growth indices (R > 0.4, P < 0.01), as does Tr (0.2 < R < 0.4, P < 0.01). The gs also exhibits a significant correlation with growth indices (0.2 < R < 0.4, P < 0.05) (Figure. 7B). The results of the RDA indicated that Pn, LMF, Tr, Chl a, and PNUE significantly influenced the growth of E. urophylla × E. grandis, explaining 50.9%, 9.6%, 4.1%, 2.8%, and 2.4%, respectively, of the variation in growth indices (Figure. 8A).

In the monoculture D. odorifera, there were significantly positive correlations between Pn, gs, Tr, CE, LNC and PNUE (0.612 < R < 0.980, P < 0.01), but all were negatively correlated with Ci and WUE (-0.849 < R < -0.446, P < 0.05). The SLA had strong negative correlations with LFM, LDM, LDMC and LTD (-0.759 < R < -0.541, P < 0.05). Furthermore, the correlation between leaf physiological and morphological traits is relatively weak and not statistically significant (-0.4 <R < 0.4, P > 0.05) (Figure. 7A). Under mixed cultivation treatment, leaf Pn, WUE, CE, and PNUE of D. odorifera are all highly significantly positively correlated (0.828 < R < 0.990, P < 0.001). The LNC and the contents of Chl a and Chl b exhibit significant correlations (R = 0.546 and 0.495, P < 0.05). The significant positive correlations are observed among LA, LV, LFM, and LDM of D. odorifera (0.831 < R < 0.944, P < 0.001). Correlations between the physiological and morphological traits of D. odorifera leaves from mixed culture were generally weak, but there were strong associations between LDMC and leaf Pn, CE, and PNUE (-0.525 < R < -0.498, P < 0.05) (Figure. 7D). In addition, Mantel's test showed significant associations between D. odorifera growth indicators and Pn, gs, Tr, WUE and PNUE in the monoculture treatment (R > 0.2, P < 0.05), while these leaf traits were only SLA, Pn and CE in the mixed cultivation treatment (0.2 <R < 0.4, P < 0.05) (Figure. 7C and D). According to the RDA results, the growth of D. odorifera was significantly affected by Pn, Tr, PNUE, and Chl a+b, which explained 55.8%, 5.3%, 2.4%, and 2.4%, respectively, of the variation in the growth indices (Figure. 8B).

Line 496: Please follow the journal guidelines for in-text citations.

Response:

Thank you, we revised it (Line 527).

Line 506: Same as above

Response:

Thank you, we revised it (Line 537).

Line 510: Same as above

Response:

Thank you, we revised it (Line 540).

Line 527: Same as above

Response:

Thank you, we revised it (Line 559). We also checked for citation problems throughout the manuscript and corrected them (Line 556).

Figure 8: Why do the authors perform RDA rather than PCA?

Overall the figures are scientifically robust and help to better understand the results.

Response: Thanks for your question. The reason we preferred to use RDA rather than PCA is that we wanted to understand how the leaf traits affected plant growth of E. urophylla × E. grandis and D. odorifera, and which leaf traits explained the largest proportion of the variation in seedling growth. With RDA, we found that Pn explained more than 50% of the variation in growth in both E. urophylla × E. grandis and D. odorifera, which was much higher than the other leaf traits. In addition, LMF, Tr, Chl a and PNUE significantly influenced growth of E. urophylla × E. grandis, whereas Tr, PNUE and Chl a+b significantly influenced growth of D. odorifera.

Line 543: “redundancy analysis (RDA) was used…” Through which software the authors perform RDA?

Response:

Thanks to your reminder, we have added software information for redundancy analysis (RDA) to the manuscript (Lines 574-575).

Line 545: “The potential correlation between leaf physiological and morphological traits was explored using correlation analysis”. Using which software please clarify.

Response:

Thanks to your reminder, we have added software information for correlation analysis to the manuscript (Line 578).

Line 562: “This study is our first attempt” Please avoid colloquial terminology.

Response: Thank you, we revised it (Line 593).

Line 572: “These research results….” Same as above

Response: Thank you, we revised it (Line 603).

Reviewer 2 Report

Comments and Suggestions for Authors

Dear Authors,

My wishes!. Good research work entitled "Leaf traits explain the growth variation and nitrogen response of Eucalyptus urophylla × eucalyptus grandis and Dalbergia odorifera in mixed culture".  However, I have some concerns, which I highlighted in the manuscript File. General comments are as follows.

The English language needs to improve in the results part. The Discussion and conclusion are okay. However, it is very hard to understand what the authors are describing. 

Kindly check the units and abbreviations and whether they are all cited properly in the text and figures. 

Some figures need high-resolution. 

All the best

Regards

Reviewer

Comments on the Quality of English Language

Author Response

Dear reviewers,

Thank you for offering us an opportunity to improve the quality of our submitted manuscript (plants-2859534). We appreciated very much the constructive and insightful comments provided by the academic editor and reviewers. In this revision, we have addressed all of these comments/suggestions. We hope the revised manuscript has now met the publication standard of your journal.

The revised portions are marked in blue ink in the paper.

Below, our point-by-point responses to the issues raised by the reviewers are listed in blue text.

Response to the comments of Reviewer #2

Dear Authors,

My wishes! Good research work entitled "Leaf traits explain the growth variation and nitrogen response of Eucalyptus urophylla × eucalyptus grandis and Dalbergia odorifera in mixed culture". However, I have some concerns, which I highlighted in the manuscript File. General comments are as follows.

The English language needs to improve in the results part. The Discussion and conclusion are okay. However, it is very hard to understand what the authors are describing.

Response: Thank you for your valuable suggestions, which will greatly improve the quality of our manuscripts. We have revised the results part overall, and hope that the new version will improve your reading experience (Lines 106-293).

Kindly check the units and abbreviations and whether they are all cited properly in the text and figures.

Response: Thank you. We have checked for formatting issues with units and abbreviations throughout the manuscript and made corrections.

Some figures need high-resolution.

Response: Thank you for your reasonable suggestions. We inserted and uploaded higher resolution (600 dpi) figures in Tiff format in the manuscript. We hope that the revised images will show the experimental results more clearly.

Specific comments:

Line 3: Use Capital

Response: Thank you, we revised it (Line 3).

Line 10: Abstract lacks methodology, include the methodology uesd

Response: Thank you for your reasonable suggestions, which greatly improved the quality of our manuscript. We have added the methodological content of this study in the abstract section (Lines 15-17).

Correction:

In this study, pot experiment was set up and seedling growth indicators, leaf physiology, morphological parameters and N content were collected and analyzed after 180 days of N application treatment.

Line 13: Use Capital

Response: Thank you, we revised it (Line 13).

Line 17: what is LNC, mention at its first introduce

Response: We used "LNC" to indicate leaf nitrogen content. We have added the full name in the abstract section (Line 20).

Line 63: use subscript

Response: Thank you, we revised it (Line 66).

Line 101: Mention these aims of the studies in abstract as well

Response: Thanks for your suggestion, we have added this aim in the abstract section (Lines 12-15).

Correction:

Thus, this study investigated the response of leaf functional traits of Eucalyptus urophylla × Eucalyptus grandis (E. urophylla × E. grandis) and Dalbergia odorifera (D. odorifera) to mixed culture and N application as well as the regulatory pathways of key traits on seedling growth.

Lines 105-108: start describing the results with Significant effects were observed in the groups, followed by the group with no effect.

Response: Thanks. We consider your suggestion to be scientific and have revised it as recommended (Lines 108-112).

Correction:

With increasing N application, there was a trend toward significantly increasing LNC, Chl a, Chl b, Chl a+b, Car, Pn, gs, photosynthetic N use sufficiency (PNUE), WUE and CE and decreasing Ci in the monoculture E. urophylla × E. grandis. Only mixed culture had no obvious effect on the LNC, Chl a, Chl b, Chl a+b, Car, gs, Tr, Ci or CE of E. urophylla × E. grandis (Figure. 1).

Line 130: submit the better resolution image

Response: Thank you. We inserted and uploaded higher resolution (600 dpi) figures in Tiff format in the manuscript. We hope that the revised images will show the experimental results more clearly (Line 136).

Lines 132-136: Mention the Figure texts (A, B.....) A is what?

Response: We apologize for the omission, and we have corrected the error by explaining the texts (A, B.....) of each little picture in the caption (Lines 137-144). We have also checked the other figures and revised them (Figure 1: Lines 128-135, Figure 3: Lines 166-170, Figure 4: Lines 172-176, Figure 5: Lines 200-206, Figure 6: Lines 208-211).

Lines 139-147: In general, the English language needs improvement. It can be difficult for readers to understand what authors are describing in their writing.

Response: We apologize for the ambiguous description. We have revised this section (Lines 146-164). Additionally, we have made comprehensive modifications to the results section to enhance your understanding of the results.

Correction:

Compared to monoculture, mixed cultivation significantly increased the LA, LDM, LDMC, and LTD of E. urophylla × E. grandis, while notably reduced the SLA (P < 0.05). On the other hand, N addition increased the LA, LDM, LDMC, and LTD of E. urophylla × E. grandis, while reducing its LT, LV, LFM, SLA, and LMF (P < 0.05). Under the combined treatment of low N addition [CK (0 g urea pot-1) and N1 (3 g urea pot-1)] and mixed cultivation, the LA, LV, LFM, LDM, LDMC, and LTD of E. urophylla × E. grandis were all higher than those under the combined treatment of low N addition (CK and N1) and monoculture. Conversely, the aforementioned traits under the combined treatment of high N addition [N2 (6 g urea pot-1) and N3 (12 g urea pot-1)] and mixed cultivation were unchanged or reduced compared to the monoculture treatment. In addition, compared to monoculture, low N addition treatments (CK and N1) reduced the SLA of E. urophylla × E. grandis leaves under mixed cultivation, while high N treatments (N2 and N3) increased the SLA.

Line 178: Check the figures., it is not the right figure, i guess.

Response: Thank you, we checked and revised it (Lines 187,188).

Line 180: shoot include leaves, leaves branches

Response: Thanks for your reasonable opinion. We believe where shoot means aboveground of seedlings, and we have added the definition and calculation of “shoot” and “root/shoot” in the methods part (Lines 550-552).

Correction:

The shoot biomass was calculated by adding the leaf, branch, and stem biomass of the seedlings, and the root/shoot ratio is the ratio of seedling root biomass to aboveground biomass.

Line 185: What does it mean when something tended to decrease and then increase? If it increases, what is the significant increase percentage?

Response: We apologize for the ambiguous description. We have revised this sentence (Lines 193-196).

Correction:

Under different nitrogen application treatments, mixed cultivation significantly (P < 0.05) reduced height, ground diameter, branch number, and biomass of leaf, rachis, stem, shoot, root, and plant of D. odorifera compared to monoculture (Figures. 6A-I).

Line 188: Is that N2 Group or N2 is means to say nitrogen?

Response: We apologize for the omission. This experiment was set up with 4 nitrogen addition treatments including no urea application (CK), 3 g urea pot-1 (N1), 6 g urea pot-1 (N2), and 12 g urea pot-1 (N3). We have added the specific methods of the experiments in the method section (Lines 499-509). And we have also explained abbreviations where it first appears in the manuscript (Lines 115-116, 123).

Line 188: maintain the N2 or N (2 with subscript)

Response: Thank you, we revised it (Lines 197). We also checked the other parts of the manuscript.

Line 190: 0.5?

Response: Thank you. It has been corrected (Lines 196-198).

Correction:

The PLER and CRDE of D. odorifera were lower than 0.5 and 1, respectively, and the N1 and N2 treatments significantly reduced the PLER and CRDE (Figures. 6K, L).

Line 196: Abbreviations should be uniform and written in subscript and regular both in legend and in text.

Response: Thank you, we have checked the formatting of all the abbreviations in the text and legends. We hope that the revised manuscript will meet with your approval.

Line 472: not necessarily to include

Response: Thank you for your reasonable suggestions and after consideration we have removed this sentence.
